# Fabrication, Characterization, and Application of Large-Scale Uniformly Hybrid Nanoparticle-Enhanced Raman Spectroscopy Substrates

**DOI:** 10.3390/mi10050282

**Published:** 2019-04-27

**Authors:** Qi Qi, Chunhui Liu, Lintao Liu, Qingyi Meng, Shuhua Wei, Anjie Ming, Jing Zhang, Yanrong Wang, Lidong Wu, Xiaoli Zhu, Feng Wei, Jiang Yan

**Affiliations:** 1School of Information science and technology, North China University of Technology, No. 5 Jinyuanzhuang Street, Shijingshan District, Beijing 100144, China; qiqi@ime.ac.cn (Q.Q.); lawliet1lch@126.com (C.L.); liulintao@ime.ac.cn (L.L.); mengqingyi@ime.ac.cn (Q.M.); zhangj@ncut.edu.cn (J.Z.); wangyanrong@ncut.edu.cn (Y.W.); yangjiang@ncut.edu.cn (J.Y.); 2State Key Laboratory of Advanced Materials for Smart Sensing, General Research Institute for Nonferrous Metals, No. 11 Xingkedong Street, Huairou District, Beijing 101402, China; weifeng@grinm.com; 3Smart Sensing R&D Center, Institute of Microelectronic of the Chinese Academy of Sciences, No. 3 Western Beitucheng Street, Chaoyang District, Beijing 100029, China; zhuxiaoli@ime.ac.cn; 4Ministry of Agriculture, Chinese Academy of Fishery Sciences, No. 150 Qingta Street, Fengtai District, Beijing 100141, China; lidongwu@mit.edu; 5Department of Chemistry, Massachusetts Institute of Technology, Cambridge, MA 02139, USA

**Keywords:** SERS, Au nanoparticles, reproducibility, sensitivity, malachite green

## Abstract

Surface-enhanced Raman spectroscopy (SERS) substrates with high sensitivity and reproducibility are highly desirable for high precision and even molecular-level detection applications. Here, large-scale uniformly hybrid nanoparticle-enhanced Raman spectroscopy (NERS) substrates with high reproducibility and controllability were developed. Using oxygen plasma treatment, large-area and uniformly rough polystyrene sphere (URPS) arrays in conjunction with 20 nm Au films (AuURPS) were fabricated for SERS substrates. Au nanoparticles and clusters covered the surface of the URPS arrays, and this increased the Raman signal. In the detection of malachite green (MG), the fabricated NERS substrates have high reproducibility and sensitivity. The enhancement factor (EF) of Au nanoparticles and clusters was simulated by finite-difference time-domain (FDTD) simulations and the EF was more than 10^4^. The measured EF of our developed substrate was more than 10^8^ with a relative standard deviation as low as 6.64%–13.84% over 15 points on the substrate. The minimum limit for the MG molecules reached 50 ng/mL. Moreover, the Raman signal had a good linear relationship with the logarithmic concentration of MG, as it ranged from 50 ng/mL to 5 μg/mL. The NERS substrates proposed in this work may serve as a promising detection scheme in chemical and biological fields.

## 1. Introduction

Surface-enhanced Raman spectroscopy (SERS), as a promising surface sensitive analytical technique, has attracted intense interest in chemical, biological, and environmental research due to its ultra-sensitivity, high selectivity, and rapid detection capability [1,2,3]. SERS provides the molecular structure and the composition of analytes by enhancing the electromagnetic field of the substrates [4,5,6]. By combining the nanostructure with a noble metal, particularly Au and Ag, the Raman scattering signal can be enhanced by several orders of magnitude due to the plasmonic resonance induced by the increase of the electromagnetic fields [7,8]. The Raman scattering light can not only be coupled with the plasmons on the surface, but also be electromagnetically enhanced when the molecules are adsorbed in hot spots [9].

However, in order to apply SERS for molecular sensing, it is important to fabricate a high-quality SERS substrate. The SERS substrate requires the characteristic of uniformity. Poor uniformity leads to the inhomogeneous distribution of hot spots, and a change in the intensity of the Raman signal. This makes it difficult to study large biomolecules [10]. The fabrication of ordered and reproducible SERS substrates has attracted intensive interest. B. Bassi reported a uniform SERS substrate made by Au nanostars grafted on functionalized glasses [11]. Kayeong Shin demonstrated that Au nanoparticle-encapsulated hydrogels could be used as a SERS substrate [12]. Wei Wu reported a reproducible SERS substrate by screen printing Ag nanoparticles on a plastic polyethylene terephthalate (PET) [13]. Liping Wu described a bio-inspired bicontinuous gyroid-structured Au SERS substrate [14]. To fabricate the ordered nano-array SERS substrates, a great number of techniques have been studied, such as photolithography [15], electron beam lithography [16], X-ray lithography [17], and nanoimprinting [18]. However, these techniques usually have shortcomings of low throughput and high cost. Meanwhile, a method based on self-assembly of polystyrene (PS) spheres presents a strategy with the advantage of low cost, high throughput, high reproducibility, and easy controllability [19]. Moreover, the uniform roughness of SERS substrates is another important characteristic. Rough SERS substrates will improve SERS activity due to the enlargement of the surface area and generation of more hot spots [20]. Among several surface roughening methods, nanosphere self-assembly has been considered as a favorable method. It was first provided by Van Duyne’s group in 2002 [21]. They developed a metal film over a nanosphere electrode as a SERS substrate. The substrates fabricated by this method are ordered and have controllable roughness, and the reproducibility of biosensors is high. ZaoYi et al. [22] and Xiaotang Hu et al. [23] reported the use of SERS substrates based on arrays of PS spheres that deposited noble metal film to detect Rhodamine 6G (R6G). The enhancement factor (EF) can reach 1.5 × 10^6^ and 10^7^ by adjusting the diameter of the PS spheres and noble metal film thickness. On the basis of their study, we used self-assembly and nanosphere lithography in conjunction with 20 nm Au films to form the nanoparticle-enhanced Raman spectroscopy (NERS) substrate. By this method, Au nanoparticles and clusters were formed on the surface of the NERS substrate, and this provided more hot spots to enhance the Raman signal. 

In practical application, SERS techniques for food safety detection have gained wide attraction. Malachite green (MG) is widely used in aquaculture because it is a highly effective parasiticide and fungicide [24]. Many studies have shown that MG has many negative effects, including high residual toxicity and carcinogenicity [25,26,27]. Due to these harmful effects, it is important to detect the drug residues of MG. SERS provides a promising way to rapidly detect MG at a low concentration. For instance, Ag nanoparticles as the SERS substrate and Ag nanoparticles decorated with silicon nanowire arrays have been used to detect MG [28,29]. Both these SERS substrates can detect MG at a low concentration, but the uniformity is poor and the distribution of the hot spots is inhomogeneous. We have focused on this problem and developed a highly reproducible substrate. 

In this study, large-scale ordered and uniform NERS substrates were fabricated by self-assembly and nanosphere lithography. The NERS substrates were made by Au sputter deposition onto uniformly rough polystyrene sphere (URPS) arrays. The Au nanoparticles and clusters covered the surface of the NERS substrates, which enhanced the Raman signal. The EF of Au nanoparticles and clusters was investigated. The measured EF of NERS substrate reached 10^8^. Finite-difference time-domain (FDTD) simulation illustrated that the EF can be increased to 10^4^ due to the Au nanoparticles and clusters.

## 2. Materials and Method

### 2.1. Materials

The suspensions (2.5 wt.%) of PS spheres (300 nm in diameter) that were used were purchased from BaseLine (Tianjin, China). Highly pure MG was purchased from Sigma-Aldrich (St. Louis, MO, USA). Sodium dodecyl sulfate (SDS, 99%), and ethanol (99.5%) were obtained from Energy-Chemical Shanghai Co., Ltd., China.

### 2.2. SERS Substrate Preparation

The PS sphere arrays were fabricated by the self-assembly technique at the air/water interface. The main steps of the process are shown in Figure 1 and include (i) filling the beaker with deionized water and adding the PS sphere suspensions mixed with 66% ethanol to the air/water interface along the beaker wall; (ii) ultrasonic concussion using 40 W power to assist the self-assembly of PS spheres; (iii) adding the proper amount of surfactant (SDS) to the water along the other side of beaker wall; and (iv) transferring the self-assembly monolayer of PS spheres onto the substrate. The URPS arrays were formed by reactive ion etching (RIE) with oxygen (O_2_) gas. The gas flow rate was 10 sccm, the radio frequency (RF) was 13.56 MHz, and the power (bottom electrode) was 40 W. Finally, a 20 nm Au film was coated by sputter deposition onto PS sphere and URPS arrays. Au has a high surface energy and adsorptive force with molecules to adsorb more molecules, so Au was used as a deposit on the surface of PS sphere and URPS arrays [30]. A scanning electron microscope (SEM, S-5500, HITACHI, Tokyo, Japan) was used to determine the characteristics of the surface of the SERS substrates and an atomic force microscope (AFM, multimode 8, Bruker, Billerica, MA, USA) was used to analyze the morphology of the SERS substrates.

### 2.3. Raman Characterization of SERS Substrate 

The Raman spectra were taken from a microconfocal laser Raman spectrometer (inVia Raman Microscope, Renishaw, Gloucestershire, UK) with an excitation wavelength at 532 nm. The diameter of the laser beam was 1 μm. SERS spectra were acquired using 50× objective with a wavenumber of 400–2000 cm^−1^. To gain a SERS signal, 6 μL of MG with different concentrations from 50 ng/mL to 5 μg/mL was dropped onto the SERS substrates and allowed to dry at ambient conditions.

### 2.4. FDTD Simulation

The model was based on the NERS substrate. The structures were illuminated with a plane wave directed along the z-axis. The periodic boundary conditions used perfectly matched layer (PML) to adsorb the boundary conditions of the cell. The simulation time was set to 300 fs. The dielectric constant for Au was selected as CRC (the type of the Au) from the database, and the refractive index of the underlying PS sphere was 1.55. The refractive index of the surrounding medium was 1.0 for air.

## 3. Results and Discussion

### 3.1. NERS Substrate Characteristics

As already mentioned, the highly ordered PS sphere arrays were fabricated by the self-assembly technique at the air/water interface. The density of the PS spheres, which was about 1.05 g/cm^3^, is slightly higher than water. Ethanol in the suspension can enhance the convection of the solvent, and the PS spheres can easily stay at the air/water interface when the suspension is injected into the water [31]. For 300 nm PS spheres, the experiments showed that a self-assembly hexagonal-close-packed (HCP) monolayer can be formed when the proportion of ethanol is 66%. A proper amount of SDS was added to promote the self-assembly performance, and this led to the increase of negative charges and counteracting van der Waals and capillary attractions [32]. In this study, the optimal condition for the formation of the ordered PS sphere arrays was a 2 mL volume of SDS. By this method, a highly ordered self-assembly monolayer can be transferred to a 2 cm × 2 cm substrate as shown in Figure 2a. It can be clearly observed from Figure 2b that the PS sphere arrays exhibit a hexagonally close-packed arrangement. The side-view SEM image in Figure 2c further demonstrates that the PS sphere arrays have a monolayer. The AFM image (Figure 2d) further illustrates a close packed hexagonal ordered arrangement of the PS sphere arrays.

The URPS arrays were formed by O_2_ plasma treatment of PS sphere arrays by using RIE with O_2_ gas. O_2_ molecules decompose into O_2_ radicals, which insert into the backbone followed by chain scission as the dominating process of etching [33]. The gas flow rate was 10 sccm, the RF was 13.56 MHz, and power was 40 W. Figure 3 illustrates the diameter of PS spheres versus etching time. The diameter diminishes linearly with the etching time until the PS spheres lose their shape (the saturation region). As the diameter of a PS sphere is reduced to approximately half its initial value, the saturated region is reached [34]. After reaching the saturation region, the curve deviates from the linear relationship. The sizes of the nanosphere arrays can be tuned by controlling the time of the etching. As shown in Figure 4a, the period can be maintained and the PS spheres can keep the spherical shape. With etching time increasing, the attacked effect becomes stronger, leading to the rough surface of the PS spheres. Moreover, the shape of the PS spheres was deformed due to anisotropic enhancement after the saturation regime. Figure 4b shows that the substrate of PS sphere arrays becomes rough after etching (compared with Figure 2d). 

The NERS substrates were processed by sputter deposition onto PS sphere arrays and URPS arrays. As shown in Figure 5a and Figure 5d, the deposition of the Au film did not disorder the structure of the PS sphere and URPS arrays. Au nanoparticles and clusters were formed on the AuURPS arrays due to RIE. However, no Au nanoparticles and clusters were formed on the surface of AuPS sphere arrays, because the surface of the PS sphere arrays was smooth. The Au nanoparticles can confine the electromagnetic hot spots in the inter-structure gaps, and this can enhance the Raman signal [35,36,37]. The Au films sputtered on the surfaces of the PS sphere arrays and URPS arrays, as shown in the side-view SEM images of Figure 5b and Figure 5e. From these observations, strong local electric fields are expected to appear at the interstitial regions and Au nanoparticles and clusters (the red region in Figure 5c and Figure 5f), which is favorable for the enhancement of the Raman signal [22]. 

### 3.2. SERS Spectra Analysis and Performance

First, the MG molecules were used to analyze the sensitivity of the NERS substrates. The chemical structure of MG is demonstrated in Figure 6a. As shown in Figure 6b, almost no signal was observed on the planar Si substrate, and very weak signals were found on the pristine PS sphere substrate. However, high Raman signals of MG were detected on the AuURPS arrays, which indicate that NERS substrates have high activity. The most prominent peaks of MG are confirmed according to the studied work [38]. The major peaks appeared at the wavenumbers of 798, 1175, 1218, 1365, 1398, and 1617 cm^−1^, and these band assignments of Raman peaks are summarized in Table 1. Most Raman signals of MG were enhanced at the excitation wavenumber of 532 nm. Figure 6c and Figure 6d demonstrate that the Raman signal of AuURPS arrays that were formed by RIE and sputtering is larger than that for the NERS substrate based on AuPS sphere arrays. This is attributable to the roughness of the substrate. The AuPS sphere arrays are smoother, which results in a limited binding capacity due to their small surface area and fewer hot spots. Furthermore, the diffusion through a stagnant layer of solvent near the surface limited adsorption kinetics from a diluted solution on a flat surface [39]. The rough AuURPS arrays have a larger surface area, which results in the adsorption of more MG molecules on the hot spots. The relationship between the roughness of SERS and Raman activity is in agreement with the electromagnetic mechanism that is the dominant mechanism to SERS enhancement. To determine the enhancement effect of AuPS sphere arrays and AuURPS arrays, the SERS EF value of MG was calculated by the following expression [40]:$$\mathrm{EF}=\frac{I_{SERS}}{I_{REF}}\cdot \frac{N_{REf}}{N_{SERS}}$$
$$N_{SERS}=\frac{C_{SERS}V_{SERS}AN_{A}}{S}\;\mathrm{and}\;N_{REF}=C_{REF}V_{REF}N_{A}$$
where *I_SERS_* is the Raman intensity of the adsorbed MG molecules on the NERS substrates, and *I_REF_* is the Raman intensity of the MG molecules on the planar Au films. *N_REF_* is the number of MG molecules excited by the laser on the surface of the planar Au film substrate. *N_SERS_* is the number of MG molecules probed on the NERS substrate, *N_A_* is Avogadro constant, *A* is the area of the laser spot (1 μm^2^), and *C_SERS_* and *C**_REF_* are the molar concentrations of MG solution in the NERS substrate and planar Au films. *V_SERS_* and *V**_REF_* are the volumes of MG solution added to the NERS substrate and planar Au films. Suppose that 6 μL 5μg/mL MG molecules are uniformly dispersed on the surface of NERS substrate and the diffusion spot is measured to be about 4 mm in diameter, so the surface area (*S*) is *S* =π(2 mm)^2^. The values of *N_SERS_* and *N_REF_* are estimated as the following:$$\begin{array}{l} N_{SERS}=\frac{5/364.9\times 10^{-3}\mathrm{mol}/L\times 6\mu L\times 1{\mu m}^{2}\times N_{A}}{\pi {\left(2\mathrm{mm}\right)}^{2}}=3.9\times 10^{6}\\ N_{REF}=5/364.9\times 10^{-3}\mathrm{mol}/L\times 6\mu L\times N_{A}=4.95\times 10^{13} \end{array}.$$

The EF value of the 1617 cm^−1^ band is calculated as follows:$$EF_{MG}(1617{\mathrm{cm}}^{-1})=\frac{3833}{229}\times \frac{4.95\times 10^{13}}{3.9\times 10^{6}}=2.1\times 10^{8}.$$

Similarly, the EF values of the AuURPS arrays substrate for 1398, 1365, 1218, 1175, and 798 cm^−1^ are estimated to be 2.57 × 10^8^, 2.19 × 10^8^, 1.57 × 10^8^, 2.14 × 10^8^, and 2.62 × 10^8^.

Next, the sensitivity of the NERS substrates was further analyzed by using a series of concentrations from 50 ng/mL to 5 μg/mL. Figure 7a and Figure 7b demonstrate that Raman intensity increases with an increase in concentration of MG with excitation wavelengths at 532 nm and 633 nm. The Raman signals show that the concentration of 50 ng/mL can be detected. The Raman signal of MG molecules at a concentration of 50 ng/mL under the 633 nm wavelength is more obvious. In particular, the three Raman peaks at 1175, 1365, and 1617 cm^−1^ exhibit distinct enhancement. Figure 7c presents the relative intensity versus the logarithms of MG concentration. The relative intensities of the characteristic Raman peaks at 1175, 1365, and 1617 cm^−1^ show a clear linear relationship with the logarithm concentration of MG. It is well known that the reproducibility and stability of Raman signals are important to the practical application of the SERS technique. The reproducibility is related to the uniform distribution of hot spots. The reproducibility is evaluated by 10 Raman spectra of the different region on one NERS substrate. A scan of 10 Raman spectra was measured spot-to-spot. As shown in Figure 7d, the 10 measurements exhibit high reproducibility. To further evaluate the SERS reproducibility, the noticeable peaks’ relative standard deviation (RSD), which is the ratio of standard deviation to the mean, was calculated from 15 spectra. Table 2 demonstrates that the RSD of six peaks are as low as ~6.64–13.84%, which shows the high reproducibility of SERS substrates. To prove that reproducibility is relative to the homogeneity of the SERS substrate, disordered AuURPS arrays were fabricated by the same method. As shown in Figure 7e, the disordered AuURPS arrays substrate presents poor reproducibility on detecting MG molecules. The RSD of disordered AuURPS arrays achieved ~50%–60%, which is five times larger than the ordered one. Therefore, the high reproducibility is attributed to the uniformity of the substrate, which is important to the application of SERS.

The electric field enhancement effect of Au nanoparticles and clusters was simulated by FDTD. The excitation wavelength of 532 nm irradiated the AuURPS from the upper side. The simulation model is shown in Figure 8a. As shown in Figure 8b, the Au nanoparticles and clusters can provide more hot spots and a maximal electric field strength (|E|/|E_0_|)^2^ of 100. For SERS intensity, it is widely believed that (|E|/|E_0_|)^4^ is the EF for an increase in Raman intensity. Thus, the Raman intensity increased to 10^4^ because of the presence of Au nanoparticles and clusters. Au nanoparticles and clusters formed by our developed method can enhance the Raman signal.

## 4. Conclusions

In conclusion, uniformly rough NERS substrates with high sensitivity and reproducibility were fabricated by coating 20 nm Au films onto URPS arrays that were formed by PS spheres using self-assembly and nanosphere lithography. The measured EF of the NERS substrates was more than 10^8^, which presents a great Raman scattering characteristic. The AuURPS arrays substrates exhibit reproducible and stable NERS signals of MG molecules with a small RSD of 6.64%–13.84% due to the spatial uniformity of the structure. The minimum detection limit was 50 ng/mL for MG molecules. The relative intensities of the characteristic Raman peaks exhibited a clear linear relationship with the logarithm concentration of MG. The FDTD simulation result confirmed that Au nanoparticles and clusters provided more hot spots and increased the EF to 10^4^. Because of these outstanding SERS characteristics, AuURPS arrays may serve as excellent NERS substrates for biomolecule detection.

## Figures and Tables

**Figure 1 micromachines-10-00282-f001:**
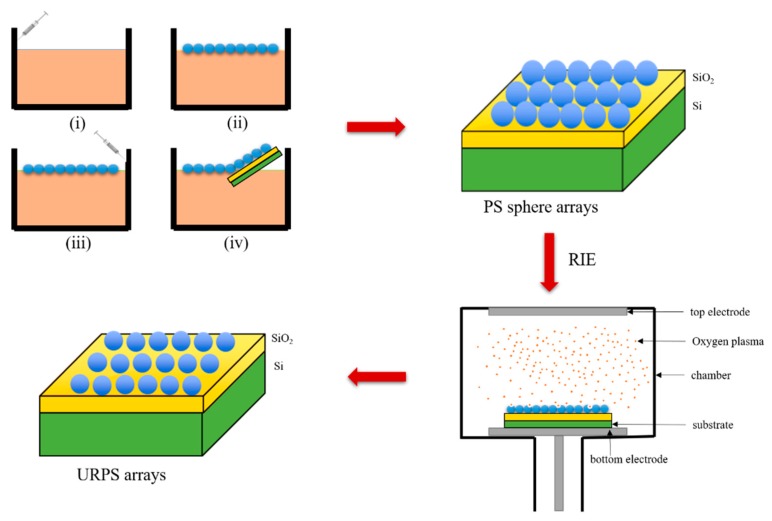
The procedure of the self-assembly monolayer of polystyrene (PS) sphere arrays and the uniformly rough polystyrene sphere (URPS) arrays. RIE: reactive ion etching.

**Figure 2 micromachines-10-00282-f002:**
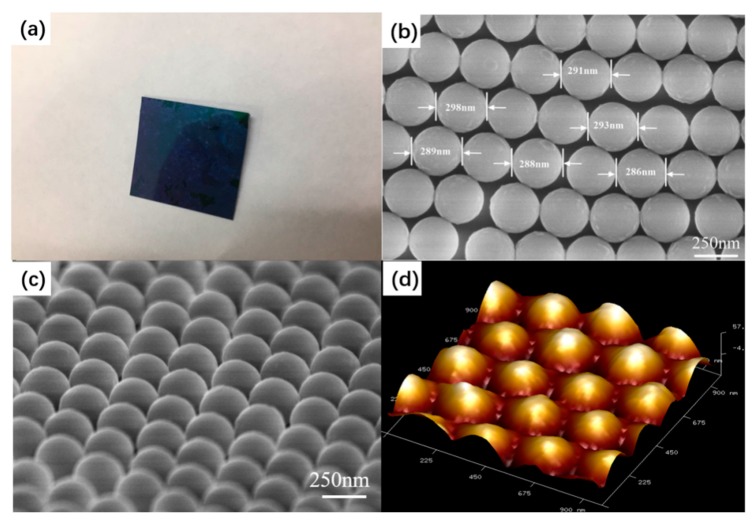
(**a**) Photograph of PS sphere arrays; (**b**) top-view SEM image of PS sphere arrays; (**c**) side-view SEM image of PS sphere arrays; and (**d**) three-dimensional (3D) atomic force microscope (AFM) image of PS sphere arrays.

**Figure 3 micromachines-10-00282-f003:**
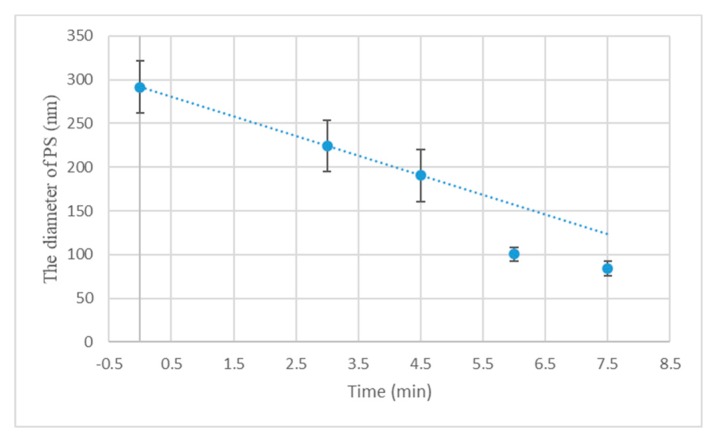
Diameter of PS spheres versus etching time.

**Figure 4 micromachines-10-00282-f004:**
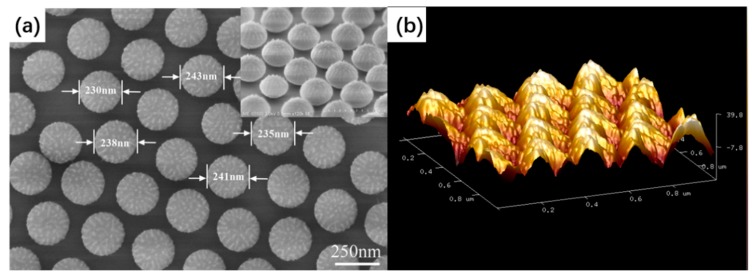
(**a**) Top-view SEM image of PS sphere arrays obtained with an etching time of 3 min; insets show the side-view SEM images of the same sample; (**b**) 3D AFM image of PS sphere arrays formed with an etching time of 3 min.

**Figure 5 micromachines-10-00282-f005:**
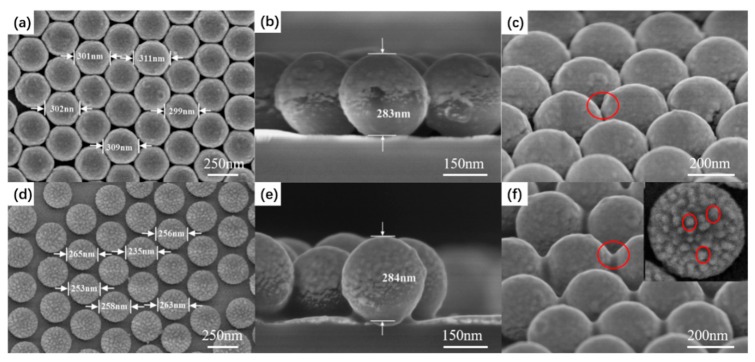
Top-view, side-view, and 30° tilted SEM image of nanoparticle-enhanced Raman spectroscopy (NERS) substrates fabricated by sputtering 20 nm Au films onto (**a**–**c**) PS sphere arrays; (**d**–**f**) PS sphere arrays after an etching time of 3 min. The inset shows the Au nanoparticles and clusters on the AuURPS arrays.

**Figure 6 micromachines-10-00282-f006:**
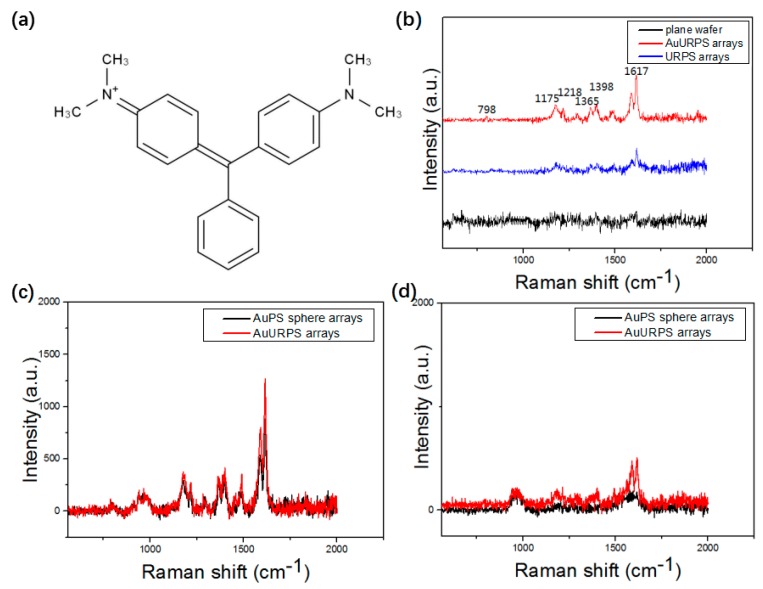
(**a**) The chemical structure of MG; (**b**) Raman signals of MG molecule (5 μg/mL) from planar Si, URPS arrays, and AuURPS arrays; Raman signal of MG molecule (**c**) 5 μg/mL; and (**d**) 500 ng/mL from AuPS sphere arrays and AuURPS arrays.

**Figure 7 micromachines-10-00282-f007:**
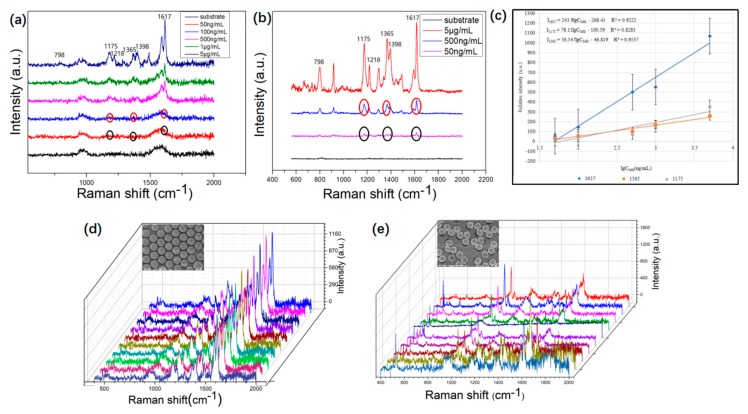
(**a**) Raman spectra of MG molecules from 50 ng/mL to 5 μg/mL under the 532 nm wavelength; (**b**) Raman spectra of MG molecules from 50 ng/mL to 5 μg/mL under the 633 nm wavelength of the same sample; (**c**) the relative intensities and linear fitting line for the Raman band at 1175, 1365, and 1617 cm^−1^ as a function of MG logarithmic concentration; (**d**) ten Raman spectra of MG molecules on the NERS substrate of ordered AuURPS arrays, and the inset shows the side-view SEM images of the ordered AuURPS arrays; and (**e**) ten Raman spectra of MG molecules on the SERS substrate of disordered AuURPS arrays, and the inset shows the side-view SEM images of the disordered AuURPS arrays.

**Figure 8 micromachines-10-00282-f008:**
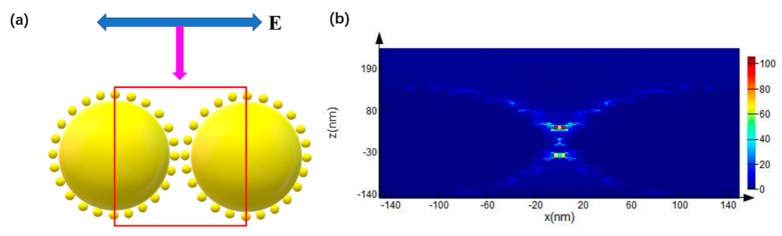
(**a**) The simulated model with Au nanoparticles and clusters; (**b**) Electric field distribution of Au nanoparticles and clusters at the excitation wavelength of 532 nm.

**Table 1 micromachines-10-00282-t001:** Assignment for surface-enhanced Raman spectroscopy (SERS) peaks of malachite green (MG) molecules.

SERS Peak Position (cm^−1^)	Assignment	Reference
798	Ring C–H out-of-plane bending (γ(C–H)_ring_)	[25,41]
1175	In-plane vibrations of ring C–H (δ(C–H)_ring_)	[42]
1218	C–H rocking (δ(C–H)_ring_)	[38]
1365	N-phenyl stretching	[43]
1398	N-phenyl stretching, δ(C–H) ring and (ν(C–C)_ring_)	[38]
1617	Ring C–C stretching (ν(C–C)_ring_)	[27]

**Table 2 micromachines-10-00282-t002:** Relative standard deviation of six noticeable MG Raman peaks from 15 Raman Spectra obtained on the ordered AuURPS arrays and disordered AuURPS arrays.

-	Wavenumber (cm^−1^)					
NERS substrate	798	1175	1218	1365	1398	1617
ordered (%)	13.84	9.48	9.99	8.27	6.86	6.64
disordered (%)	50.2	38.52	33.65	50.17	58.26	63.77

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
