# Peer review of "Fabrication, Characterization, and Application of Large-Scale Uniformly Hybrid Nanoparticle-Enhanced Raman Spectroscopy Substrates"

_micromachines, 2019, doi:10.3390/mi10050282_

Round 1
Reviewer 1 Report
Dear Editor,
I would recommend its publication after the following minor revisions:
1) Some shortcuts have been defined twice and some once. This should be standardized.
2) The readability of the manuscript should be improved, e.g. sentence in line 62: "Many researches have been reported before." Please explain what they presented. Additionally, only two literature references refer to this part of the manuscript. If the authors use the term "Many" then there should be more references.
3) Fig. 3 - how did the authors assume that 4.5 minutes is a limit value? For 5 measuring points made with an accuracy of 1.5 minutes, it is difficult to deduce a limit value with an accuracy of 0.1 minutes.
4) The description of the sampling procedure does not contain information on the number of samples. The method of checking the repeatability should be described in the manuscript.
5) Fig. 7 - missing descriptions of individual charts (a, b, c and d)
Author Response
1. Some shortcuts have been defined twice and some once. This should be standardized.
Authors’ response: We have modified this mistake and given the shortcuts in the first appearance.
2. The readability of the manuscript should be improved, e.g. sentence in line 62: "Many researches have been reported before." Please explain what they presented. Additionally, only two literature references refer to this part of the manuscript. If the authors use the term "Many" then there should be more references.
Authors’ response: We have deleted this sentence “Many researches have been reported before.” The main purpose of giving these two literature references is to explain the reported SERS substrates structure for detecting the MG and emphasize the importance of uniformity.
3. Fig. 3 - how did the authors assume that 4.5 minutes is a limit value? For 5 measuring points made with an accuracy of 1.5 minutes, it is difficult to deduce a limit value with an accuracy of 0.1 minutes.
Authors’ response: The 4.5 minutes is not the limit value. The saturated region can be reached when the diameter of PS is reduced to approximate half of its initial value. In this experiment the saturated region is after 4.5 min. It has been revised in this manuscript. (line 167-168)
4. The description of the sampling procedure does not contain information on the number of samples. The method of checking the repeatability should be described in the manuscript.
Authors’ response: In this manuscript, the reproducibility is evaluated by 10 Raman spectra of the different region on one NERS substrate sample as shown in Figure 7d. It has been emphasized on the manuscript. (line 252-253)
5. Fig. 7 - missing descriptions of individual charts (a, b, c and d).
Authors’ response: It has been revised in the manuscript as shown in Figure 7.
Reviewer 2 Report
The paper deals with a very interesting and hot topic.
The proposed strategy to obtain reproducible and efficient SERS substrates sounds very interesting. Nevertheless the quality of the language, a lot of misprints, a coincise report of the results and some missing information lead to a difficult for the reader to obtain a satisfactory point of view matching strategy and results of the research. In my opinion the matter is interesting but the paper must be deeply improved before pubblication.
In the following some minor and major (in bold) remarks:
Line 19: ….substrates with high sensitivity and reproducibility is highly desirable.... ARE!
Lines 27-28: apex style should be applied
Line 29: Moreover, the Raman signal of the SERS substrates has a good linear relationship with the logarithmic concentration of MG from 50 ng/mL to 5 μg/mL.
NO! It is the signal from MG not “of the substrates”.
Concerning the use of acronym NERS: is it possible to enhance non resonant Raman signal without nanoparticles?
Line 40-42: “several orders of magnitude due to the plasma resonance induced the electro-magnetic fields increase”. I think that “by” is missing. It is better to speak about plasmonic resonance instead plasma
Line 42-44: “The Raman scattering light can be interconnected with the plasmons on the surface when the molecules adsorbed in the hot spots, and the electromagnetic field can be enhanced [9].” The sentence is uncomplete and in this form not valid.
Line 45: “The SERS substrate requires the characteristic with uniformity.“
The term “roughed” don't exist
In Figure 1 the red arrow in Figure 1b indicate a structure in which the difference from Figure 1a is only due to a lower amount of PS. Is it correct? Please explain better in the text.
What is RIE? Reactive Ion Etching? Please expand it in the first appearance.
Scale and labels in figure 2, 4 and 5 are unreadable.
It is better to avoid “intensive Raman Signal”, maybe high Raman signal
“The mainly peaks”??
“these band”
“The EF values of the AuURPS arrays substrate for the 1617, 1398, 1365, 1218, 1175 and 798 cm-1 are estimated to be 4.7×109, 6.54×108, 9.65×108, 7.37×108 and 2.15×109.”
The mentioned peaks are six; the reported EF values are 5.
In order to evaluate Ef values it is important to quantify “NSERS... the number of MG molecules probed on the NERS substrate.” and explain how it is obtained.
Line 227: what means CRC?
The scale in Figura 6 c must be modified putting zero at the bottom of the panel.
The references recalled in introduction speaking about the state of the art about reproducible and stable SERS substrates must be enlarged at least with the following items:
K.Shin et al “Au nanoparticle-encapsulated hydrogel substrates for robust and reproducible SERS measurement” 932 | Analyst, 2013, 138, 932–938
B. Bassi et al. “Robust, reproducible, recyclable SERS substrates: monolayers of gold nanostars grafted on glass and coated with a thin silica layer.” Nanotechnology. 2019 Jan 11;30(2):025302. doi: 10.1088/1361-6528/aae9b3.
L. Wu et al. “Highly sensitive, reproducible and uniform SERS substrates with a high density of three-dimensionally distributed hotspots: gyroid-structured Au periodic metallic materials “, NPG Asia Materials (2018) 10, e462; doi:10.1038/am.2017.230
T. Lee et al, “Highly robust, uniform and ultra-sensitive surface- enhanced Raman scattering substrates for microRNA detection fabricated by using silver nanostructures grown in gold nanobowls” Nanoscale, 2018, 10, 3680–3687
W. Wu et al., “Low-Cost, Disposable, Flexible and Highly Reproducible Screen Printed SERS Substrates for the Detection of Various Chemicals” Scientific RepoRts | 5:10208 | DOi: 10.1038/srep10208
Author Response
1. Line 19: ….substrates with high sensitivity and reproducibility is highly desirable.... ARE!;
Lines 27-28: apex style should be applied;
Line 29: Moreover, the Raman signal of the SERS substrates has a good linear relationship with the logarithmic concentration of MG from 50 ng/mL to 5 μg/mL. NO! It is the signal from MG not “of the substrates”.
Authors’ response: They have been revised in the manuscript. (line20, 28-29, 31)
2. Concerning the use of acronym NERS: is it possible to enhance non resonant Raman signal without nanoparticles?
Authors’ response: NERS represents nanoparticles-enhanced Raman spectroscopy. In our work, the uniformly hybrid substrate included the PS spheres and Au nanoparticles which is formed by RIE and sputtering. We compared the AuPS (only PS spheres) arrays with AuURPS arrays (PS spheres and Au nanoparticles). The results indicated the Raman intensity of AuURPS arrays was higher than AuPS arrays due to the Au nanoparticles. It has been explained in the manuscript. (line 185-188)
3. Line 40-42: “several orders of magnitude due to the plasma resonance induced the electro-magnetic fields increase”. I think that “by” is missing. It is better to speak about plasmonic resonance instead plasma;
Line 42-44: “The Raman scattering light can be interconnected with the plasmons on the surface when the molecules adsorbed in the hot spots, and the electromagnetic field can be enhanced [9].” The sentence is uncomplete and in this form not valid;
Line 45: “The SERS substrate requires the characteristic with uniformity. The term “roughed” don't exist.
Authors’ response: These mistakes have been revised in the manuscript. (line 43-45)
4. Figure 1 the red arrow in Figure 1b indicate a structure in which the difference from Figure 1a is only due to a lower amount of PS. Is it correct? Please explain better in the text.
Authors’ response: These diagrams represented the substrate before and after RIE respectively. We have revised the Figure 1 as shown in manuscript.
5.What is RIE? Reactive Ion Etching? Please expand it in the first appearance.
Authors’ response: Yes. RIE is the Reactive Ion Etching. It has been expanded in the first appearance. (line 118)
6. Scale and labels in figure 2, 4 and 5 are unreadable.
Authors’ response: We have added the readable scale and labels according to the SEM resolution in the manuscript as shown in Figure 2, 4, 5.
7. The EF values of the AuURPS arrays substrate for the 1617, 1398, 1365, 1218, 1175 and 798 cm-1 are estimated to be 4.7×109, 6.54×108, 9.65×108, 7.37×108 and 2.15×109.” The mentioned peaks are six; the reported EF values are 5. In order to evaluate Ef values it is important to quantify “NSERS... the number of MG molecules probed on the NERS substrate.” and explain how it is obtained.
Authors’ response: We have given the expression of EF, NSERS and NREFto calculate the EF value of MG. It has been explained in the manuscript. (line 218-235)
8.Line 227: what means CRC?
Authors’ response: It is the type of the Au in the FDTD simulation. (Line 140)
9. The scale in Figure 6 c must be modified putting zero at the bottom of the panel.
Authors’ response: The minimum value of the vertical coordinates has been changed. It has been revised in the manuscript as shown in Figure 6.
10. The references recalled in introduction speaking about the state of the art about reproducible and stable SERS substrates must be enlarged at least with the following items:
Authors’ response: These references have been added. [11-14,18]
[11]. B. Bassi et al. Robust, reproducible, recyclable SERS substrates: monolayers of gold nanostars grafted on glass and coated with a thin silica layer. Nanotechnology 2019, 30.
[12]. K.Shin et al. Au nanoparticle-encapsulated hydrogel substrates for robust and reproducible SERS measurement. Analyst 2013, 138, 932–938.
[13]. W. Wu et al. Low-Cost, Disposable, Flexible and Highly Reproducible Screen Printed SERS Substrates for the Detection of Various Chemicals. Scientific RepoRts 2015.
[14]. T. Lee et al. Highly robust, uniform and ultra-sensitive surface- enhanced Raman scattering substrates for microRNA detection fabricated by using silver nanostructures grown in gold nanobowls. Nanoscale 2018, 10, 3680-3687.
[18]. L. Wu, et al. Highly sensitive, reproducible and uniform SERS substrates with a high density of three-dimensionally distributed hotspots: gyroid-structured Au periodic metallic materials. NPG Asia Materials 2018, 10.
Reviewer 3 Report
Micromachines-478308
The paper reports an interesting work on SERS detection of Malachite green using a metal film over spheres substrate. The methodology of the preparation of such SERS substrates is not new and one of the firsts works reporting the use of metal film over spheres platforms as SERS substrates was in 2002. Van Duyne’s group is pioneer in these kind of SERS substrates, however the authors did not reference any work of this group. I suggest the authors to read “Materials Today, 15, 2012, 16-25” and “J. Phys. Chem. B, 2002, 106 (4), pp 853–860”
I suggest that the authors try to revise the manuscript by explaining their distinction and the originality of their work. In general, the manuscript fits very well into the field covered by the journal Micromachines. However, the current state of the manuscript is below the journal standard. Consequently, I recommend publication in Micromachines after major revision. In addition, few comments might help to address some open questions, see below.
1. In the introduction, the authors claim “The enhancement factor can be reached to
69 1.5×106 by adjusting the diameter of the PS spheres and Ag film thickness. Xiaotang Hu et al. reported a SERS substrate based on PS spheres array, in which deposited Au film, Au nanoparticles and clusters cover the surface of the PS spheres array by adjusting the thickness of the Au film. The enhancement factor was greater than 107.” What was the molecular probe used in these works? Malachite green? The molecular probe has a huge influence on the EF.
2. In the materials and method, the authors should indicate the full name, supply and purity for SDS, ethanol, Au NPs or electrode, water (was distilled or ultra pure), O2etc. What are RIE and RF?
3. Page 6, The authors claimed “The EF values of the AuURPS arrays substrate for the1617, 1398, 1365, 1218, 1175 and 798 cm-1 are estimated to be 4.7×109, 6.54×108, 9.65×108, 7.37×108 and 2.15×109.” Why the authors calculate the EF for the more intense bands of MG? Normally the EF is calculated for one band. Why this variation of the EF for the different bands?
4. The chemical structure of MG should be indicated in Figure 6
5. Figure 7-a: I do not see any characteristic Raman bands for MG in the Raman spectra for 50ng/mL and 100ng/mL, even with the circles highlighted by the authors. This should be revised.
6. Figure 7-b. The points should have standard deviation. How many Raman spectra the authors have taken to produce Figure 7-b?
7. The FDTD calculation should be explained in more detail in the Materials and Methods section. At least the programme used and the model.
Author Response
I suggest that the authors try to revise the manuscript by explaining their distinction and the originality of their work. In general, the manuscript fits very well into the field covered by the journal Micromachines. However, the current state of the manuscript is below the journal standard. Consequently, I recommend publication in Micromachines after major revision. In addition, few comments might help to address some open questions, see below.
Authors’ response: The difference between their and our work is that because we use reactive ion etching to treat PS spheres surface, the PS sphere surface became rough (URPS arrays). After sputtering Au over the URPS arrays surface, Au nanoparticles and clusters were formed on URPS surface, which make the Raman signal enhanced. It has been explained in the manuscript. (line 67-69,73-76,97-103)
1. In the introduction, the authors claim “The enhancement factor can be reached to 69 1.5×106 by adjusting the diameter of the PS spheres and Ag film thickness. Xiaotang Hu et al. reported a SERS substrate based on PS spheres array, in which deposited Au film, Au nanoparticles and clusters cover the surface of the PS spheres array by adjusting the thickness of the Au film. The enhancement factor was greater than 107.” What was the molecular probe used in these works? Malachite green? The molecular probe has a huge influence on the EF.
Authors’ response: The molecular probe used was Rhodamine 6G (R6G) in their works. It has been added in the manuscript. (line 69-71)
2. In the materials and method, the authors should indicate the full name, supply and purity for SDS, ethanol, Au NPs or electrode, water (was distilled or ultra pure)(LINE 90), O2etc. What are RIE and RF?
Authors’ response: The full names of SDS, RIE and RF have been explained in the manuscript. Sodium dodecyl sulfate (SDS, 99%), and ethanol (ethanol, 99.5%) were obtained from Energy- Chemical Shanghai Co., Ltd., China. This information has been added in the materials and method. (line 106-109)
3. Page 6, The authors claimed “The EF values of the AuURPS arrays substrate for the1617, 1398, 1365, 1218, 1175 and 798 cm-1 are estimated to be 4.7×109, 6.54×108, 9.65×108, 7.37×108 and 2.15×109.” Why the authors calculate the EF for the more intense bands of MG? Normally the EF is calculated for one band. Why this variation of the EF for the different bands?
Authors’ response: In the following two references, they also calculated the EF for the more intense bands.
[1].X Hu, Z Xu, K Li, F Fang, L Wang. Fabrication of a Au–polystyrene sphere substrate with three-dimensional nanofeatures for surface-enhanced Raman spectroscopy. Applied Surface Science 2015, 355, 1168–1174.
[2]. Tong Yang, et al. Hydrogen-Bond-Mediated in Situ Fabrication of AgNPs/Agar/PAN Electrospun Nanofibers as Reproducible SERS Substrates. ACS Appl. Mater. Interfaces 2015, 7, 3, 1586-1594.
4. The chemical structure of MG should be indicated in Figure 6
Authors’ response: The chemical structure of MG has been shown in Figure 6a.
5. Figure 7-a: I do not see any characteristic Raman bands for MG in the Raman spectra for 50ng/mL and 100ng/mL, even with the circles highlighted by the authors. This should be revised.
Authors’ response: We supplemented a set of experimental data. The Raman spectra of MG molecules at concentration of 50ng/mL can be detected under the 633 nm wavelength as shown in Figure 7b.
6. Figure 7-b. The points should have standard deviation. How many Raman spectra the authors have taken to produce Figure 7-b?
Authors’ response: The revised figure was shown in the manuscript. We calculated the average of the intensity from 8 Raman spectra to produce Figure 7c.
7. The FDTD calculation should be explained in more detail in the Materials and Methods section. At least the programme used and the model.
Authors’ response: It has been explained in the Materials and Methods section. (line136-140). The model was based on the NERS substrate. The structures were illuminated with a plane wave directed along the z-axis. The periodic boundary conditions used perfectly matched layer (PML) adsorbing boundary conditions of the cell. The simulation time was set 300fs. The dielectric constant for Au was selected as CRC from the database. And the refractive index of underlying PS sphere was 1.55. The refractive index of surrounding medium was 1.0 for air.
Round 2
Reviewer 2 Report
The authors have improved the paper and answered to the remarks of the referees.
Only two weak remarks:
- in figure 2b and in figure 4a the SEM images are presented with diameters written with white characters. I guess that in the present form that numbers prove to be invisible.
- in figure 3 the PS diameters should be presented with proper error bars.
Thus I think the paper can be accepted after considering the above reported minor remarks.
Author Response
Reviewer #2 comment:
1.in figure 2b and in figure 4a the SEM images are presented with diameters written with white characters. I guess that in the present form that numbers prove to be invisible.
Authors’ response:We have amplified the white characters of the PS diameters in the SEM images as shown infigure 2, figure 4 and figure 5
2.in figure 3 the PS diameters should be presented with proper error bars..
Authors’ response:We have added the error bars in the figure 3.
Reviewer 3 Report
The manuscript can be accept for publication in Micromachines in present form.
Author Response
Thank you for your advising.
